



# A quality-assured dataset of nine radiation components observed at the Shangdianzi regional GAW station in China (2013−2022)

Weijun Quan[1,2], Zhenfa Wang[3], Lin Qiao[1], Xiangdong Zheng[4], Junli Jin[5],
Yinruo Li[1], Xiaomei Yin[1], Zhiqiang Ma[3], and Martin Wild[2]

[1]Beijing Weather Forecast Centre, Beijing Meteorological Service, Beijing, 100097, China
[2]Institute for Atmospheric and Climate Science, ETH Zürich, 8092 Zürich, Switzerland
[3]Institue of Urban Meteorology, China Meteorological Administration, Beijing, 100089, China
[4]Chinese Academy of Meteorological Science, China Meteorological Administration, Beijing, 100081, China
[5]Meteorological Observation Centre, China Meteorological Administration, Beijing, 100081, China

Correspondence: Weijun Quan (quanquan78430@163.com) and Martin Wild (martin.wild@env.ethz.ch)

**Abstract.** A New Baseline Surface Radiation (NBSR) system was established at the Shangdianzi (SDZ) regional Global Atmosphere Watch (GAW) station in 2013 to observe nine broadband radiation components, i.e., the global-, direct-, diffuse-, and upwelling-shortwave irradiance (GSWI, DSWI, DifSWI, and UpSWI) as well as the photosynthetically active radiation (PAR), ultraviolet irradiance (UVAI; UVBI), down- and up-welling long-wave irradiance (DnLWI; UpLWI). To test the 1-min raw radiometric data, a Hybrid Algorithm for Radiation Data Quality Control (HARDQC) is presented in this study based on well-established methods together with the solar irradiance dataset and the spectral features of the instrument bands. Subsequently, an NBSR dataset, which consists of multi-time scale (i.e. 1-min, hourly, daily, monthly, monthly average hourly, and monthly average daily) radiation datasets over 2013–2022, is established and evaluated. Results show that more than 98.7% of all radiation components passed the physical possibility test. The percentages passed the extremely rare test are greater than 98.6% for all radiation components except for the DnLWI (97.1%). The percentages passed the comparison test are greater than 83.3% (GSWI), 78.3% (DSWI), 81.7% (DifSWI), 93.1% (UpSWI), 88.9% (PARI), 95.6% (UVAI), 96.3% (UVBI), 99.8% (DnLWI), and 99.7% (UpLWI), respectively. Due to data logger faults, removal of the instruments for calibration, and lightning strokes, some apparent data gaps in the upwelling radiation components (January 2015–August 2017) and all radiation components (December 2018; July to September 2021) were detected. Despite the existence of a few imperfections in the NBSR dataset, it is still reliable to apply in many fields such as the validation of satellite products and numerical models, investigation of relationships between radiation and atmospheric composition, and the detection of changes in the surface fluxes.





# 1 Nomenclature

$I_{sc}$    Solar constant, i.e., extra-terrestrial irradiance over whole spectrum incident on a unit area exposed normally to rays of the sun at one astronomical unit (1367 W m$^{-2}$)

$E_0$    Eccentricity correction factor of the Earth's orbit

$\theta_z$    Solar zenith angle (degree)

$\mu = \cos\theta_z$   Cosine of solar zenith angle

$\sigma = 5.67 \times 10^{-8}$ W m$^{-2}$ K$^{-4}$    Stephan-Boltzman constant

P    Barometric pressure (hPa)

$T_a$    Air temperature in Kelvin

RH    Relative humidity (%)

Vis    Visibility (km)

$W_s$    Wind speed (m s$^{-1}$)

$W_d$    Wind direction (°)

$I_g$    Global shortwave irradiance (GSWI) incident upon a horizontal surface (W m$^{-2}$)

$I_{nb}$    Direct shortwave irradiance normal to the solar rays (DnSWI) (W m$^{-2}$)

$I_b = I_{nb} \cdot \cos\theta_z$   Direct (or beam) shortwave irradiance (DSWI) incident upon a horizontal surface (W m$^{-2}$)

$I_d$    Diffuse shortwave irradiance (DifSWI), i.e., the component of global radiation that is scattered out of the solar beam by the atmospheric constituents (W m$^{-2}$)

$I_r$    Upwelling shortwave irradiance (UpSWI), i.e., the part of the global radiation that is reflected by the surface (W m$^{-2}$)

$q_{par}$    Downwelling photosynthetically active radiation quantum (PAR) on a horizontal surface in the spectral interval 400–700 nm (µmol s$^{-1}$ m$^{-2}$)

$I_{par}$    Downwelling photosynthetically active radiation irradiance (PARI) on a horizontal surface in the spectral interval 400–700 nm (W m$^{-2}$)

$I_{uva}$    Downwelling irradiance on a horizontal surface in the spectral interval 315–400 nm (UVAI; W m$^{-2}$)

$I_{uvb}$    Downwelling irradiance on a horizontal surface in the spectral interval 280–315 nm (UVBI; W m$^{-2}$)

$I_{dl}$    Downwelling long-wave irradiance (DnLWI) on a horizontal surface, which is the thermal emission of the atmosphere incident on the planet's surface (W m$^{-2}$)

$I_{ul}$    Upwelling terrestrial long-wave irradiance (UpLWI) from a horizontal surface, i.e., the thermal emission from the planet's surface (W m$^{-2}$)

$I_{ns} = I_g - I_r$   Net solar irradiance (NSRI) on a horizontal surface (W m$^{-2}$)

$I_{nl} = I_{dl} - I_{ul}$   Net long-wave irradiance (NLRI) on a horizontal surface (W m$^{-2}$)

$I_{\mathrm{nt}} = I_{\mathrm{ns}} + I_{\mathrm{nl}}$ Net irradiance (NTRI) on a horizontal surface (W m$^{-2}$)

$K_{\mathrm{t}} = I_{\mathrm{g}}/(I_{\mathrm{sc}} \cdot E_0)$ Clear index, which is the percentage attenuation by the atmosphere of the incoming global radiation

$K_{\mathrm{d}} = I_{\mathrm{d}}/I_{\mathrm{g}}$ Diffuse coefficient, which reflects the capability of atmosphere to diffuse the incoming radiation

$K_{\mathrm{u}} = I_{\mathrm{uva}}/I_{\mathrm{uvb}}$ Ratio of UVA to UVB, which is usually assigned to assess the effects of ozone on ultraviolet radiation

$\alpha = I_{\mathrm{r}}/I_{\mathrm{g}}$ Albedo, which is the ratio of UpSWI to GSWI, is considered as an essential parameter of surface energy budget.

## 2 Introduction

High-temporal-resolution ground based radiation measurement plays a critical role in lots of applications such as verifying satellite retrievals (e.g., Yan et al., 2011; Zhang et al., 2013) and numerical model predictions (e.g., Nowak et al., 2008; Garcá et al., 2014, 2018; Gueymard et al., 2016), establishing parameterization models (e.g., Yang et al., 2023), understanding of the genesis and evolution of Earth's climate (e.g., Ohmura et al., 1998; Deriemel et al., 2018), investigating global dimming/brightening (e.g., Wild et al., 2005, 2009), assessing renewable energy technologies (e.g., Yang et al., 2022),

as well as evaluating the influence of atmospheric components on energy budget (e.g., Garcá et al., 2008; Mol et al., 2023). Whereas, it is a hard task to accurately measure the high-temporal-resolution radiation components due to several constraints, e.g. the malfunction of the instruments, inappropriate operating by the operators, inadequate calibration of the instruments, lack of quality control (QC) approaches, as well as the influence of the adverse weather conditions (Ohmura et al., 1998; Younes et al., 2005; Journé and Bertrand, 2011;Vuilleumier et al., 2014). To support the research projects of the World

Climate Research Programme (WCRP) and other scientific programs, the World Meteorological Organization/International Council of Scientific Unions (WMO/ICSU) joint Scientific Committee for the WCRP proposed the establishments of the international Baseline Surface Radiation Network (BSRN) in October 1988 (Ohmura et al., 1998; Driemel et al., 2018). The BSRN network was designed to offer a long-term high temporal resolution (1 min) Earth's surface irradiance to validate satellite-based estimates of the surface radiation budget, to verify and improve radiation codes of climate models, as well as

to monitor long-term changes in irradiance at the Earth's surface. The accuracy targets for BSRN shortwave irradiance measurements are 0.5% for direct normal and 2% for global or diffuse irradiance (McArthur, 2005). The BSRN operation started in 1992 at nine sites (Ohmura et al., 1998), and now consists of 76 stations (51 active, 9 declared as inactive, 16 are closed in 2023) situated in various climatic zones, covering a latitude range from 80° N to 90° S (https://bsrn.awi.de/nc/stations/maps/, last access: 25 April 2023). As a part of the BSRN, the Surface Radiation Budget

Network (SURFRAD), which was established by United States in 1993, has provided abundant high-temporal-resolution data of broadband solar and thermal infrared irradiance for satellite retrieval and modelling validation, climate and hydrology research since 1995 (e.g., Augustine et al., 2000; Yang and Boland, 2019). At present, seven SURFRAD stations are operating in climatologically diverse regions: Montana, Colorado, Illinois, Mississippi, Pennsylvania, Nevada, and South



Dakota (https://gml.noaa.gov/grad/surfrad/overview.html, last access: 26 April 2023). Meanwhile, the U.S. Atmospheric
Radiation Measurement program (ARM) is another famous network for monitoring surface radiation with the most rigorous
procedures (e.g., Stokes and Schwartz, 1994; Ackerman and Stokes, 2003; Peppler et al., 2008).

In China, high-temporal-resolution (1 min or 3 min) radiation measurements merely date back to 1994 (Quan et al.,
2021). Under the auspice of the Global Environment Facility of the United Nations Development Programme (UNDP-GEF),
the China Meteorological Administration (CMA) established the first high-temporal radiation observations at the Mt.
Waliguan (WLG) Global Atmosphere Watch (GAW) station in 1994. Up to now, the WLG has yielded a long-term
(1994−2019) high-temporal-resolution dataset of GSWI, DifSWI, DnSWI, and near-infrared solar irradiance (NirSWI) (e.g.,
Zhou, 2005; Wei, 2010; Quan et al., 2021). Apart from the WLG, three regional GAW stations of the CMA, i.e., the
Shangdianzi (SDZ), Lin'an, and Longfengshan stations, began to perform high-resolution (1-min) observations of both short-
and long-wave radiation in 2005 (e.g., Song 2013; Quan et al., 2021). Furthermore, the CMA has established a network
entitled China Baseline Surface Radiation Network (CBSRN) since 2013, which currently consists of seven stations, i.e.
Mohe, Xilinhot, Yanqi, SDZ, Xuchang, Wenjiang, and Dali (e.g., Li et al., 2013; Yang et al., 2023). Apart from the CMA,
the Chinese Academy of Sciences (CAS) has established a Chinese Ecosystem Research Network (CERN) since 2004, in
which both the short- and long-wave radiation flux were recorded in 1-min resolution at 44 stations (e.g., Hu et al., 2007a,
2007b, 2012; Liao et al., 2020; http://cern.ac.cn/1wljs/index.asp, last access: 15 May 2023). Especially, as a unique BSRN
station in China, Xianghe (XH), which is operated by the Institute of Atmospheric Physics of CAS, has measured the
radiation components since approximately fifteen years (e.g., Driemel et al., 2018; Liu et al., 2023). The long-term
measurements at XH have been applied in many scientific fields (e.g., Yu, 2006; Yu et al., 2008; Bai et al., 2018; Liu et al.,
2021, 2022) as well as yielded a high-temporal-resolution (1 min) database of short- and long-wave radiation (Liu et al.,
2023). In addition, some high-resolution scientific research radiation stations were established by universities in China. For
instance, the Semi-Arid Climate and Environment Observatory of Lanzhou University (SACOL) station (35 ′34′N, 104 ′5′E;
1965.8 m a.s.l.) and the Wuhan University station (30°32′N, 114 ′21′E; 30 m a.s.l.) stations, which were established by the
Lanzhou university and Wuhan university, respectively, have a capability to provide high-temporal-resolution measurements
of both short- and long-wave radiation (e.g., Huang et al., 2008; Wang et al., 2014).

As one of the earliest regional GAW stations of the WMO in China, SDZ has been observing upward and downward
shortwave and long-wave broadband irradiances in 1-min resolution by means of two radiation observation systems since
2005. One is the so-called Old Baseline Surface Radiation (OBSR) system, which is mainly composed of the precision
radiometers manufactured by the Eppley laboratory (e.g. the PSP pyranometer, the 8-48 pyranometer, the PIR pyrgeometer,
and the NIP pyrheliometer) as well as the UV radiometer made by the Kipp & Zonen (e.g. the UV-S-B-T radiometer). The
other is entitled New Baseline Surface Radiation (NBSR) system, which consists of precision radiometers made by Kipp &
Zonen (e.g. the CMP11 pyranometer, the CHP1 pyrheliometer, the UV-S-AB-T radiometer), the Hukseflux (e.g. the IR02
pyrgeometer), and the Campbell Scientific Inc. (e.g. the Li190SB Quantum sensor). The NBSR system has been established
to measure nine radiation components (i.e. GSWI, DnSWI, DifSWI, UVAI, UVBI, PAR, UpSWI, DnLWI, and UpLWI)



since 2013. The OBSR system had been introduced by the CMA to observe seven radiation components (i.e. GSWI, DnSWI, DifSWI, UVBI, UpSWI, DnLWI, and UpLWI) at SDZ during 2005–2016. Therefore, it is probable to improve the consistency of long-term radiation data at the SDZ by comparing the simultaneous measurements from both the OBSR and the NBSR systems during the overlapping period (2013–2016). However, besides some previous investigations relevant for the measurements of the OBSR (Cheng et al., 2009; Quan et al., 2009a; Quan et al., 2010; Song, 2013), few studies have been carried out with respect to the observations of the NBSR. The aim of this study, thus, is to foster the utilization of the NBSR measurements in both scientific research field and operational applications via establishing a quality-assured dataset of nine radiation components at SDZ. This paper is organized as follows: the site and the associated instruments are depicted in Section 3. Section 4 describes the associated theoretical calculations and the QC procedures. Detail descriptions of the establishment, the evaluation, and the potential application of the NBSR dataset are conducted in the Section 5. Finally, Section 6 is dedicated to the summary and discussions.

## 3  Site and measurements

### 3.1 Site description

SDZ (40.65 °N, 117.12 °E; 293.3 m a.s.l.) is located in the centre of the Beijing−Tian−Hebei (BTH) region and south of the *Yanshan* Mountains, approximately 150 km northeast to downtown Beijing. Some medium-sized cities (e.g., Zhangjiakou, Chengde, Tangshan, Langfang, and Baoding) and the Megacity Tianjin are circulated around it, and to its south east is the *Bohai Sea* (Fig. 1a). The working area of SDZ is situated on the top of a hill in Miyun county of Beijing, which consists of a gradient observation tower (GOT; denoted with "A" in Fig. 1b), a vertical sounding area (VSA; denoted with "B" in Fig. 1b), a meteorological observation field (MOF; denoted with "C" in Fig. 1b), and a two-story detection and office building (DOB; denoted with "D" in Fig. 1d). Note that the measurements of meteorological elements as well as the UpSWI and UpLWI are conducted in the MOF, while the GSWI, DnSWI, DifSWI, PARI, UVAI, UVBI, and DnLWI are performed using the radiometers mounted on two brackets at the rooftop of the DOB. The MOF is located to the northwest of the DOB and has approximately the same elevation as the top of the DOB, which warrants the measurements for downward- and upward radiation at the same level.

Major land cover types of SDZ including conifer forest, chestnut trees, pear trees, maize, etc. Only small villages in mountainous areas have sparse population within 30 km of SDZ. The anthropogenic emission, thus, is negligible and the measurements at SDZ can be considered as representative for the background status of the atmospheric components over North China. SDZ was initially established by the CMA as a meteorological observation station in 1958, and has become one of the earliest regional GAW stations of WMO in China in 1981 (Zhou et al., 2021).



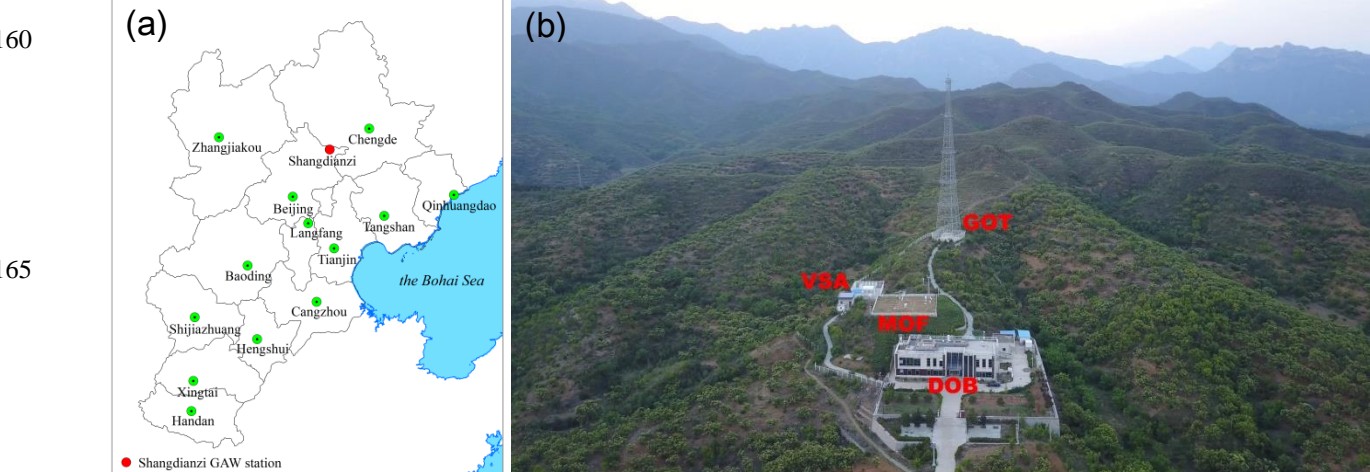

**Figure 1.** Geographic location of SDZ in BTH (a), and a drone photo of the working area of SDZ (b), which consists of the GOT, the VSA, the MOF, and the DOB.

## 3.2 Irradiance measurements

### 3.2.1 Instruments and data storage

Figure 2 and Table 1 summarize the general characteristics of the instruments in the NBSR at SDZ. The DnSWI is measured by a CHP1 (Kipp & Zonen, the Netherlands) pyrheliometer (denoted with '1' in Fig. 2a), which is installed on a two-axis automatic sun tracker (FT-ST22) manufactured by the Jiangsu Radio Scientific Institute Co., Ltd., China. The FT-ST22 is mounted on a 1.5 m high off-white fiberglass grating on the rooftop of the DOB. The CHP1 is a widely used high-accuracy radiometer (response time < 5s; non-linearity <0.5 % per year) to measure direct normal solar irradiance in the range of 200–4000 nm (Table 1).

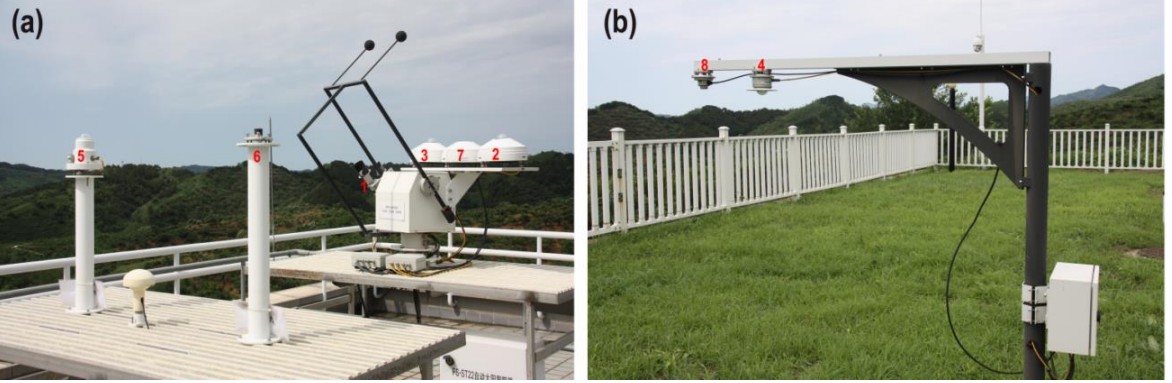

**Figure 2.** Photos of the downward radiation measuring platform located on the rooftop of the DOB at SDZ (a), and the upward radiation measuring bracket rooted in the MOF (b). The CHP1 (#120933; #190707) pyrheliometer (denoted with "1") is selected to measure the DnSWI; The face-up CMP11 (#115978; #163082)



pyranometer (denoted with "2") with shading and ventilation is used to measure the DifSWI; The face-up
CMP11 (#115977; #184975) without shading but with ventilation (denoted with "3") is utilized to observe the
GSWI; The face-down CMP11 (#127676; #185056) with a glare screen kit (denoted with "4") is adopted to
observe the UpSWI; Both the UVAI and the UVBI are monitored by a UV-S-AB-T (#120109) broadband UV
radiometer (denoted with "5"); The Li190SB quantum sensor (denoted with "6") is used to observe the PAR;
The face-up IR02 (#1277; #2033) pyrgeometer with shading and ventilation (denoted with "7") is designated
to observe the DnLWI; The face-down IR02 (#1267; #1679) without ventilation (denoted with "8") is adopted
to measure the UpLWI.

One face-up CMP11 (Kipp & Zonen, the Netherlands) pyranometer mounted on the FT-ST22 (denoted with '3' in Fig.
2a) is designed to measure the GSWI, while the other with a shade ball is used to record the DifSWI (denoted with '2' in Fig.
2a). To improve the reliability and accuracy of the radiation measurements, a ventilation and heating device is attached on
the CMP11, which can alleviate the thermal offsets as well as automatically clean any dirt that may fall on the dome of the
pyranometer such as snow, frost, precipitation, dust, and air pollutants. On the other hand, a face-down CMP11 (denoted
with '4' in Fig. 2b) installed on the top of the 1.5 m high steel bracket on the MOF is allotted to measure the UpSWI. It has
no ventilation and heating device but has an accessory glare screen kit, which protect the pyranometer dome from being
illuminated by the direct solar irradiance at low solar elevation (e.g., < 5 °) (Ohmura et al., 1998; Kipp & Zonen, 2016). The
CMP11 meets the criterion of the ISO-9060 secondary standard pyranometer with a spectral range of 285–2800 nm (Table 1).
Moreover, it has several new features and benefits such as low dome thermal offset error ($\leqslant$ ±2 W m$^{-2}$ at 5K/h of
temperature change), excellent cosine/directional response ($\leqslant$ ±10 W m$^{-2}$), outstanding long term stability of sensitivity, and
excellent linearity (non-linearity $\leqslant$ ±0.6 %) performance (Kipp & Zonen, 2016).

The DnLWI and UpLWI are measured by IR02 pyrgeometers (Hukseflux, the Netherlands) with a spectral range of
4.5–42 μm, which covers most of the spectrum of atmospheric long-wave radiation. The face-up IR02 (#1277; #2033),
which is mounted on the FS-ST22 automatic solar tracker and shaded with a shade ball (denoted with '7' in Fig. 2a), is used
to measure the DnLWI. The shade ball prevents excessive heating of the dome by the sun and shields the sensor from
receiving the infrared fraction of the solar beam (e.g., Philipona et al., 1998; Driemel et al., 2018), which is not part of the
atmospheric long-wave radiation (e.g., Enz et al., 1975). The temperature dependence of IR02 is within ±3 % (−10 to 40 ℃),
and a ventilation/heating system is installed beneath it to alleviate the influence of environmental temperature as well as to
avoid dew/dust fall on its window. The face-down IR02 (#1267; #1679) installed on the bracket (denoted with '8' in Fig. 2b)
is used to measure the UpLWI. Note that the field of view of the IR02 pyrgeometer is 150 ° rather than the desired 180 °,
which decreases its price while keeping a relatively high accuracy (Hukseflux, 2022).

Besides the radiation components related to the surface radiation budget, other ancillary radiation components (e.g., the
PARI, the UVAI, and the UVBI) are also monitored in this station. A dual-band ultraviolet radiometer UV-S-AB-T (Kipp &
Zonen, 2012), which is mounted on the top of a 1.2 meter high steel column on the other off-white fiberglass grating
(denoted with '5' in Fig. 2a), is employed to simultaneously measure the ultraviolet radiation both in the spectra of UVA



(315–400 nm) and UVB (280–315 nm). Meanwhile, a Li190SB quantum sensor (Campbell Scientific, Inc., U.S.A.) installed on the top of a 1.5 meter high steel column on the same grating (denoted with '6' in Fig. 2a) is adopted to monitor the

220 Photosynthetic Photon Flux Density (PPFD) in $\mu$mol s$^{-1}$ m$^{-2}$ (CSI, 2015). The PPFD reflects the number of photons in the 400–700 nm waveband received during a given time on a unit surface (Hu et al., 2007a).

On the other hand, a total of six proxy instruments (i.e., #190707, #163082, #184975, #185056, #2033, and #1679) were used to substitute their counterparts (i.e., #120933, #115978, #115977, #127676, #1277, and #1267) to measure the DnSWI, DifSWI, GSWI, UpSWI, DnLWI, and UpLWI after 9 September 2020 (Table 1).

**Table 1.** Brief summary of the instruments employed to measure nine radiation components at SDZ.

| No. | Component | Serial No. | Observing period | Type | Spectral range (nm) | Instrument | Manufacturer |
|---|---|---|---|---|---|---|---|
| 1 | $I_{nb}$ | #120933 | 20130101−20200908 | CHP1 | 200–4000 | Pyrheliometer | Kipp&Zonen the Netherlands |
| | | #190707 | 20200909− Present | | | | |
| 2 | $I_d$ | #115978 | 20130101−20200908 | CMP11 | 285–2800 | Pyranometer | |
| | | #163082 | 20200909− Present | | | | |
| 3 | $I_g$ | #115977 | 20130101−20200908 | | | | |
| | | #184975 | 20200909− Present | | | | |
| 4 | $I_r$ | #127676 | 20130101−20200908 | | | | |
| | | #185056 | 20200909− Present | | | | |
| 5 | $I_{uva}$; $I_{uvb}$ | #120109 | 20130101− Present | UV-S-AB-T | 315–400; 280–315 | UV radiometer | |
| 6 | $q_{par}$ | #46893 | 20130101− Present | Li190SB | 400–700 | Quantum sensor | Campbell Sci. Inc. U.S.A |
| 7 | $I_{dl}$ | #1277 | 20130101−20200908 | IR02 | 4500–40000 | Pyrgeometer | Hukseflux the Netherlands |
| | | #2033 | 20200909− Present | | | | |
| 8 | $I_{ul}$ | #1267 | 20130101−20200908 | | | | |
| | | #1679 | 20200909− Present | | | | |

All radiation components mentioned in Table 1 are sampled every second, and 1-min statistics (i.e., the mean, minimum, maximum, and sample standard deviation) are recorded as a so-called station-to-archive file in the data logger of the NBSR (Jiangsu Radio Scientific Institute Co., Ltd., China). The relatively complex, strictly defined format of the station-to-archive file is intended to encapsulate both the data and all relevant metadata within a single ASCII file. The first record of the file is

230 the fundamental information relevant for the observation including the station identifier, observation date, longitude, latitude, and elevation of the station, the installing heights and the identifiers of the instruments, etc. The actual data is stored starting with the second record in the file, in which the byte order of items is listed in Table 2.

**Table 2.** Byte order of the actual data record in the station-to-archive file.

| No. | Item | Byte | Unit | No. | Item | Byte | Unit | No. | Variable | Byte | Unit |
|---|---|---|---|---|---|---|---|---|---|---|---|
| 1 | YYYY-MM-DD hh:mm[*] | 16 | | 16 | ventilating speed of pyranometer for $I_g$ | 4 | m s$^{-1}$ | 31 | stdev of $I_{ul}$ | 8 | W m$^{-2}$ |
| 2 | mean of $I_{nb}$ | 4 | W m$^{-2}$ | 17 | body temp of | 4 | ℃ | 32 | cavity temp of | 4 | ℃ |





| | | | | | | pyranometer for $I_g$ | | | | | pyrgeometer for $I_{ul}$ | | |
|---|---|---|---|---|---|---|---|---|---|---|---|---|---|
| 3 | min of $I_{nb}$ | 4 | W m$^{-2}$ | 18 | mean of $I_r$ | 4 | W m$^{-2}$ | 33 | mean of $I_{uva}$ | 4 | W m$^{-2}$ |
| 4 | max of $I_{nb}$ | 4 | W m$^{-2}$ | 19 | min of $I_r$ | 4 | W m$^{-2}$ | 34 | min of $I_{uva}$ | 4 | W m$^{-2}$ |
| 5 | stdev of $I_{nb}$ | 8 | W m$^{-2}$ | 20 | max of $I_r$ | 4 | W m$^{-2}$ | 35 | max of $I_{uva}$ | 4 | W m$^{-2}$ |
| 6 | mean of $I_d$ | 4 | W m$^{-2}$ | 21 | stdev of $I_r$ | 8 | W m$^{-2}$ | 36 | stdev of $I_{uva}$ | 8 | W m$^{-2}$ |
| 7 | min of $I_d$ | 4 | W m$^{-2}$ | 22 | mean of $I_{dl}$ | 4 | W m$^{-2}$ | 37 | mean of $I_{uvb}$ | 4 | W m$^{-2}$ |
| 8 | max of $I_d$ | 4 | W m$^{-2}$ | 23 | min of $I_{dl}$ | 4 | W m$^{-2}$ | 38 | min of $I_{uvb}$ | 4 | W m$^{-2}$ |
| 9 | stdev of $I_d$ | 8 | W m$^{-2}$ | 24 | max of $I_{dl}$ | 4 | W m$^{-2}$ | 39 | max of $I_{uvb}$ | 4 | W m$^{-2}$ |
| 10 | ventilating speed of pyranometer for $I_d$ | 4 | m s$^{-1}$ | 25 | stdev of $I_{dl}$ | 8 | W m$^{-2}$ | 40 | stdev of $I_{uvb}$ | 8 | W m$^{-2}$ |
| 11 | body temp of pyranometer for $I_d$ | 4 | °C | 26 | ventilating speed of pyrgeometer for $I_{dl}$ | 4 | m s$^{-1}$ | 41 | temp of UV radiometer | 4 | °C |
| 12 | mean of $I_g$ | 4 | W m$^{-2}$ | 27 | cavity temp of pyrgeometer for $I_{dl}$ | 4 | °C | 42 | mean of $q_{par}$ | 4 | μmol s$^{-1}$ m$^{-2}$ |
| 13 | min of $I_g$ | 4 | W m$^{-2}$ | 28 | mean of $I_{ul}$ | 4 | W m$^{-2}$ | 43 | min of $q_{par}$ | 4 | μmol s$^{-1}$ m$^{-2}$ |
| 14 | max of $I_g$ | 4 | W m$^{-2}$ | 29 | min of $I_{ul}$ | 4 | W m$^{-2}$ | 44 | max of $q_{par}$ | 4 | μmol s$^{-1}$ m$^{-2}$ |
| 15 | stdev of $I_g$ | 8 | W m$^{-2}$ | 30 | max of $I_{ul}$ | 4 | W m$^{-2}$ | 45 | stdev of $q_{par}$ | 8 | μmol s$^{-1}$ m$^{-2}$ |

*True solar time

### 3.2.2 Instrument maintenance and calibration

All instruments adopted in this study are carefully maintained by the operators at the SDZ, which includes regular cleaning of dust and snow deposited on the dome, replacing the desiccant in the instruments, etc. In order to assure the radiation measurements at the SDZ to be traceable to the World Radiometric Reference, all instruments employed in the NBSR were sent to the manufacturer (Jiangsu Radio Scientific Institute Co., Ltd.) to be calibrated against the reference instruments (e.g. the CM21 pyranometer, the CHP1 pyrheliometer, the CGR4 pyrgeometer, the UVS-AB-T radiometer, and the Li-200190SB sensor). These reference instruments had been compared against the national radiometric standards of China (e.g. the CM22 pyranometer, the H-F absolute cavity radiometer, the CG4 pyrgeometer, etc.), which were transferred from the World Radiation Center in Davos, Switzerland (e.g., Quan et al., 2010; Yang et al., 2015; PMOD/WRC, 2022; Yang et al., 2023).

Table 3 summarizes the instruments employed in the NBSR, the sensitivities of instruments, and the ratio of changes of sensitivities between two calibration campaigns. The original sensitivities of the instruments were supplied by the manufacturer before the installation in the NBSR. Subsequently, the first calibration of the instruments was performed by the manufacturer on 14 November 2018. It can be seen from Table 3 that the sensitivities of the pyrheliometer (#120933), the pyranometers (#115978, #115977, and #127676), and the pyrgeometers (#1277 and #1267) are relatively stable over the whole observation period, e.g., the ratio of change ranges from −0.4% to 0.8%. Nevertheless, the sensitivities of the UVS-AB-T radiometer (#120109) altered apparently, i.e., the ratios of sensitivity changes are 10.7% and 14.7% in the UVA and UVB band, respectively. The Li190SB quantum sensor (#46893) had a moderate sensitivity change ratio of −3.0%.

Earth System
Science
Data

Open Access | Discussions

Additionally, six new instruments, which includes the pyrheliometer (#190707), the pyranometers (#163082, #184975, and #185056), and the pyrgeometers (#2033 and #1679), were utilized to replace their counterparts as listed in Table 3 on 19 August 2020.

**Table 3.** The sensitivities of the instruments employed in the NBSR at SDZ.

| No. | Radiation Component | Instrument type | Instrument ID | Instrument calibration | | | | |
|-----|-----|-----|-----|-----|-----|-----|-----|-----|
| | | | | Sensitivity | Calibration date | Sensitivity | Calibration date | Ratio of change (%) |
| 1 | $I_{nb}$ | CHP1 | #120933 | 7.76 | 11 Jun 2012 | 7.76 | 14 Nov 2018 | 0.0 |
| | | | #190707 | 8.45 | 20 Jul 2020 | | | − |
| 2 | $I_d$ | CMP11 | #115978 | 8.78 | 09 Dec 2011 | 8.80 | 14 Nov 2018 | 0.2 |
| | | | #163082 | 9.17 | 20 Jul 2020 | | | − |
| 3 | $I_g$ | | #115977 | 9.00 | 09 Dec 2011 | 8.96 | 14 Nov 2018 | −0.4 |
| | | | #184975 | 9.12 | 20 Jul 2020 | | | − |
| 4 | $I_r$ | | #127676 | 9.16 | 19 Jul 2012 | 9.14 | 14 Nov 2018 | −0.2 |
| | | | #185056 | 9.09 | 20 Jul 2020 | | | − |
| 5 | $I_{uva}$ | UV-S-AB-T | #120109 | 30.01 | 17 Aug 2012 | 33.23 | 14 Nov 2018 | 10.7 |
| 6 | $I_{uvb}$ | | | 2.04 | 17 Aug 2012 | 2.34 | 14 Nov 2018 | 14.7 |
| 7 | $q_{par}$ | Li190SB | #46893 | 3.98[*] | 17 Aug 2012 | 3.86[*] | 14 Nov 2018 | −3.0 |
| 8 | $I_{dl}$ | IR02 | #1277 | 10.34 | 31 Jul 2012 | 10.40 | 14 Nov 2018 | 0.6 |
| | | | #2033 | 11.64 | 19 Jul 2020 | | | − |
| 9 | $I_{ul}$ | | #1267 | 10.98 | 23 Jul 2012 | 11.07 | 14 Nov 2018 | 0.8 |
| | | | #1679 | 12.81 | 19 Jul 2020 | | | − |

The unit of sensitivity marked with the symbol * is μv (μmol s$^{-1}$ m$^{-2}$)$^{-1}$, otherwise the unit is μv (W m$^{-2}$)$^{-1}$.

### 3.3 Meteorological elements measurements

Fundamental meteorological elements (e.g. near surface air temperature, pressure, wind direction/speed, relative humidity, and visibility) are also measured at SDZ using an automatic weather station (AWS) in a 1-min interval and stored in a
260 HY3000 data logger (Huayun Sounding Meteorological Technology Inc., China). The air temperature and relative humidity are detected by a humidity and temperature probe HMP 115 (Vaisala, Finland) mounted within an instrument shelter. Barometric pressure is detected via a PTB110 barometer (Vaisala, Finland) and the wind speed/direction is monitored by a cup anemometer mounted on the top of 10 m high tower in the MOF. A DNQ1 visibility meter (Huayun Sounding



Meteorological Technology Inc., China) rooted on the MOF is adopted to measure the visibility (Li et al., 2021). The raw

data of the meteorological elements is processed carefully by experienced engineers according to the standards of WMO to

yield a high-quality meteorological dataset.

# 4 Quality control of basic radiation components

Apart from six radiation components (i.e., the GSWI, the DifSWI, the DnSWI, the UpSWI, the DnLWI, and the UpLWI)

that can be quality controlled using the recommended methods of the BSRN (e.g., Long and Dutton, 2002; Long and Shi,

2008), the three other components (i.e., the PAR, the UVAI, and the UVBI) are also included in the NBSR. Therefore, a

comprehensive QC scheme appropriate for all nine radiation components is presented in this study as follows.

## 4.1 Unit conversion of PAR observations

PAR is the solar radiation received at the Earth's surface in the visible electromagnetic spectrum (400–700 nm), and it can

be expressed in energy units (W m$^{-2}$) or in quantum units (μmol s$^{-1}$ m$^{-2}$) depending on the actual application (Hu et al.,

2007). In other words, the former is adopted when the energy aspect of radiation is required while the latter is mostly applied

in photobiology. As the photon energy is inversely proportional to its wavelength, the number of photons should decrease

with the reduction of the wavelength in the case of radiative energy conservation. Thus, the ideal spectral response function

of the quantum sensor should increase with wavelength rather than keep a constant value (denoted with the dashed lines in

the Fig. 3a). The actual spectral response function of the Li190SB (denoted with the thick curve in the Fig. 3a) is fairly close

to that of the ideal quantum response function with a systematic spectral error not exceeding 1% (Ross and Sulev, 2000; CSI,

2015).

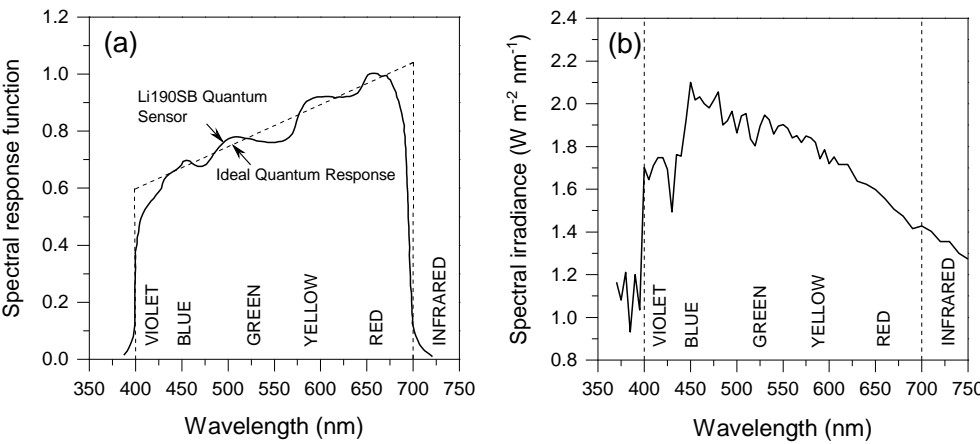

**Figure 3.** Spectral response function of the Li190SB quantum sensor (a), and spectral irradiance in the
spectral interval 400−700 nm (b). Data source of spectral irradiance: WRC spectrum (Iqbal, 1983).





The raw data of PAR observed by the Li190SB of NBSR at SDZ is recorded in the quantum units ($\mu$mol s$^{-1}$ m$^{-2}$), which can be converted to energy units (W m$^{-2}$) according to the formula presented by Nie et al. (1992).

$$I_{\text{par}} = 6.02 \times 10^{17} hc \frac{\int_{400}^{700} I_\lambda d\lambda}{\int_{400}^{700} \lambda I_\lambda d\lambda} \times q_{\text{par}} = 119.7 \frac{\int_{400}^{700} I_\lambda d\lambda}{\int_{400}^{700} \lambda I_\lambda d\lambda} \times q_{\text{par}} \,, \tag{1a}$$

where $I_{\text{par}}$ is the PARI (W m$^{-2}$) at the land surface; $q_{\text{par}}$ is the PAR directly measured by the Li190SB ($\mu$mol s$^{-1}$ m$^{-2}$); $h$ is Planck's constant ($h = 6.626 \times 10^{-34}$ J s), $c$ is velocity of light ($c = 3.0 \times 10^{17}$ nm s$^{-1}$); $I_\lambda$ is the spectral irradiance, which was derived from the WRC solar spectral irradiance dataset (Iqbal, 1983) and adopted in this study (see Fig. 3b) ; $\lambda$ is the wavelength in nm. After an integral calculation of the spectral irradiance, Eq. 1a is simplified as:

$$I_{\text{par}} = 0.220 \times q_{\text{par}} \,, \tag{1b}$$

## 4.2 Calculation of eccentricity and solar zenith angle

During QC processing of the shortwave irradiance, two astronomical parameters (i.e. $\theta_z$ and $E_0$) are required to provide the essential geographic location of the sun. $E_0$ can be estimated in terms of the day number of the year by using the formula presented by the Spencer (1971).

$$E_0 = 1.000110 + 0.034221 \cos \Gamma + 0.001280 \sin \Gamma + 0.000719 \cos 2\Gamma + 0.000077 \sin 2\Gamma \,, \tag{2}$$

In this equation, $\Gamma$, in radians, is called the day angle. It is represented by

$$\Gamma = 2\pi(d_n - 1)/365 \,, \tag{3}$$

where $d_n$ is the day number of the year, ranging from 1 on 1 January to 365 on 31 December (Iqbal, 1983).

For a given geographical position, in the absence of the Earth's refractive atmosphere, the trigonometric relations between the sun and a horizontal surface are well known as follows (Iqbal, 1983):

$$\theta_z = \cos^{-1}[\sin \delta \sin \varphi + \cos \delta \cos \varphi \cos \omega] \,, \tag{4}$$

where $\theta_z$ is the zenith angle in degrees; $\varphi$ is the geographic latitude of the site in degrees, north positive; $\delta$ is the declination, the angular position of the sun at solar noon with respect to the plane of the equator, north positive, in degrees. Spencer (1971) presented the following expression for $\delta$, in degrees:

$$\delta = (0.006918 - 0.399912 \cos \Gamma + 0.070257 \sin \Gamma - 0.006758 \cos 2\Gamma \\ + 0.000907 \sin 2\Gamma - 0.002697 \cos 3\Gamma + 0.00148 \sin 3\Gamma)(180/\pi) \,, \tag{5}$$

This equation estimates $\delta$ with a maximum error $< 3'$.

The $\omega$ in equation (4) is the hour angle, with noon zero and morning positive, which can be expressed in hours as follows (Iqbal, 1983):

$$\omega = \cos^{-1}[-\tan \varphi \tan \delta] \,. \tag{6}$$



## 4.3 Processing workflow of quality control

The QC of radiometric observations involves several tests, in which the measured irradiance is checked whether it passes the test by comparing it to the limits (Journée and Bertrand, 2011). Though the NBSR is well maintained and all instruments are regularly calibrated, some irrational records still exist due to the influence of adverse weather, operational mistakes, power failure, data transmission interrupt, etc. Therefore, the QC procedure is essential in improving the quality of baseline surface radiation data as well as its potential application. To this end, a Hybrid Algorithm for Radiation Data Quality Control (HARDQC) is presented in this study to test the 1-min raw data of nine radiation components observed by the NBSR. Several previous QC approaches (e.g. Papaioannou et al., 1993; Long and Dutton, 2002; CMA, 2007; Hu et al., 2007a, 2007b; Long and Shi, 2008) as well as the new presented limits, which were derived from a calculation based on the solar spectral irradiance dataset and the spectral response functions of the instruments, are incorporated in the HARDQC algorithm to fulfil the QC procedure for all nine radiation components. If a radiation record is detected by the HARDQC as a suspicious value, it is marked with a relevant flag but not deleted, which provides an option for the individual NBSR customers to decide how to effectively use the data depending on the aim of their work. Moreover, the QC procedure can also deliver necessary information for the operator to solve the related problems in the NBSR during the operation.

Figure 4 exhibits the flow chart of the QC algorithm for the1-min raw data of the NBSR. First, the observation date, the latitude and longitude of SDZ, shortwave irradiance, and long-wave irradiance are extracted from the original record of the NBSR. Second, the eccentricity and solar zenith angle are calculated in terms of the observation time, the latitude and the longitude of the site by using the empirical formulae (Iqbal, 1983). Third, the PARI is calculated from the PAR using the Eq. 1b. Hereto, seven shortwave radiation components and two long-wave radiation components have been processed to perform the QC algorithm. Fourth, the "Physically possible limits test" is conducted on the basis of the input shortwave and long-wave irradiances. The QC flag is set to 6/5 if the measured irradiance exceeds the upper/lower limit in this step, otherwise, the procedure will proceed to the next step. Fifth, if the irradiance that has passed the "physically possible limits test" fails to pass the "Extremely rare limits test", then, the QC flag will be set to 4 (greater than the upper limit) or 3 (less than the lower limit), otherwise, it enters into the next step. Sixth, if the irradiance that has passed the previous tests does not pass the "Comparison test between radiation components", then the QC flag is set to 2 (greater than the upper limit) or 1(less than the lower limit). Eventually, if the irradiance has passed all three tests, the associated QC flag is set to 0, which represents the best quality data recording in the NBSR. Note that, the air temperature is also involved in the "Comparison test between radiation components" to assess the QC of the long-wave irradiance.

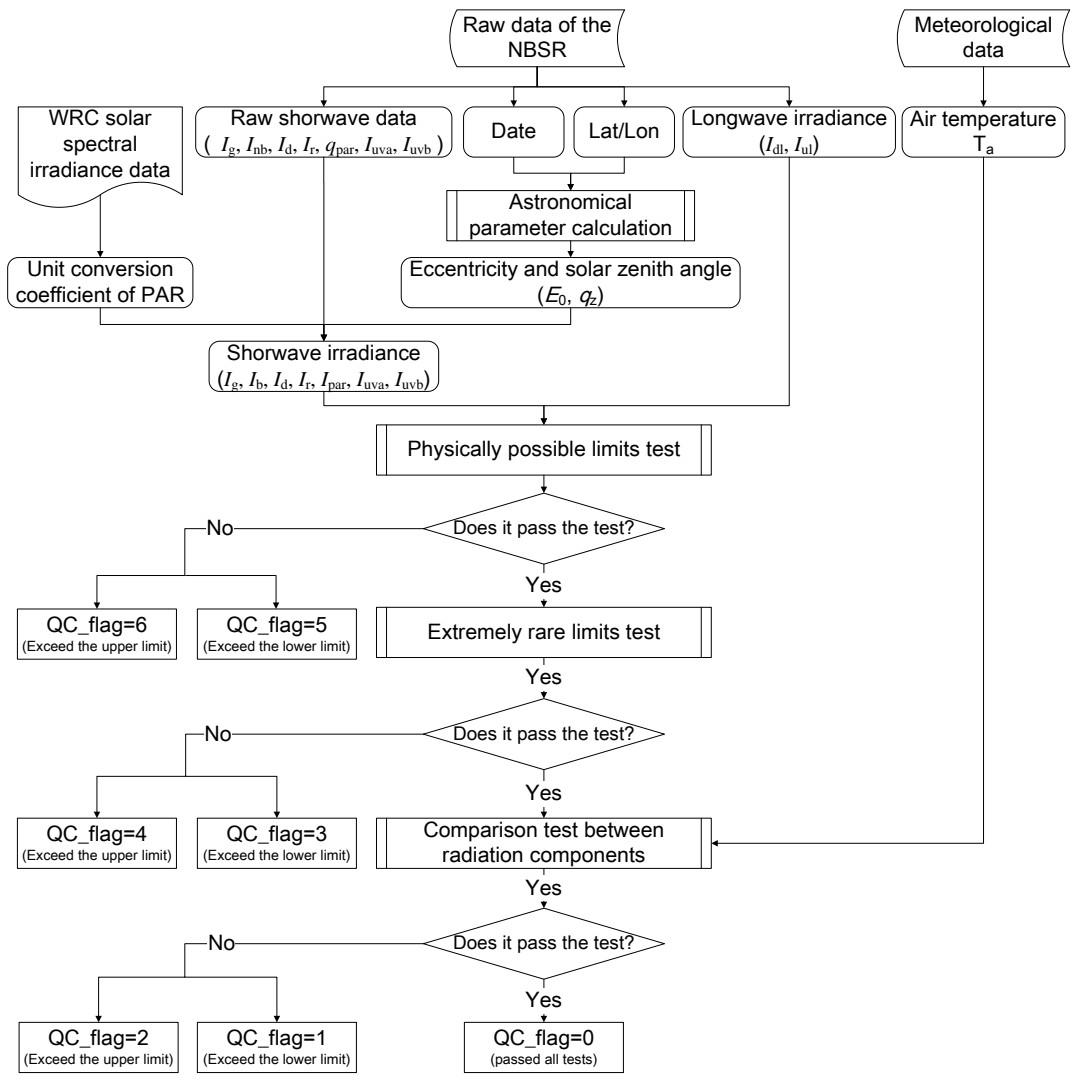

**Figure 4.** Flow chart of the HARDQC procedure based on the 1-min raw data observed by the NBSR.

### 4.3.1 Physically possible limits test

This is the first step of the HARDQC procedure conducted to detect extremely large observational errors and random errors

associated with the data handling (e.g., Ohmura et al., 1998; Long and Shi, 2008). In this test, the irradiance that falls outside

the limit is assigned a QC flag of 6 (above the maximum limit) or 5 (below the minimum limit). The lower and upper limits

for nine radiation components in the "Physically possible limits text" recommended in this study are summarized in Table 4.

**Table 4.** Lower and upper limits in the "Possible physical limits test".

| Lower limit (W m$^{-2}$) | Radiation component | Upper limit (W m$^{-2}$) | Source |
|---|---|---|---|
| −4 | $I_g$ | $I_{sc} \times E_0 \times 1.5 \times \mu^{1.2} + 100$ | Long and Shi, 2008 |



| Lower limit (W m$^{-2}$) | Radiation component | Upper limit (W m$^{-2}$) | Source |
|---|---|---|---|
| −4 | $I_b$ | $I_{sc} \times E_0 \times \mu$ | |
| −4 | $I_d$ | $I_{sc} \times E_0 \times 0.95 \times \mu^{1.2} + 50$ | |
| −4 | $I_r$ | $I_{sc} \times E_0 \times 1.2 \times \mu^{1.2} + 50$ | |
| 40 | $I_{dl}$ | 700 | |
| 40 | $I_{ul}$ | 900 | |
| −2 | $I_{par}$ | $0.39 \times I_{sc} \times E_0 \times 1.5 \times \mu^{1.2} + 39$ | This work |
| −1 | $I_{uva}$ | $0.06 \times I_{sc} \times E_0 \times 1.5 \times \mu^{1.2} + 6$ | |
| −1 | $I_{uvb}$ | $0.01 \times I_{sc} \times E_0 \times 1.5 \times \mu^{1.2} + 1$ | |

Table 4 shows that the limits for the GSWI, the DSWI, the DifSWI, the UpSWI, the DnLWI, and the UpLWI in this
study are same as those in QC of the BSRN data recommended by Long and Shi (2008), but the limits for the PARI, the
UVAI, and the UVBI were derived by scaling the solar constants over the spectrum of the PAR (533.7 W m$^{-2}$), UVA (85.0
W m$^{-2}$), and UVB (18.6 W m$^{-2}$) to that over the whole spectrum (1367.0 W m$^{-2}$). Namely, the corresponding coefficients of
0.39, 0.06, and 0.01 as well as the intercepts of 39, 6, and 1 W m$^{-2}$ were adopted for modulating the upper limits of the PARI,
the UVAI, and the UVBI, respectively. It is worth noting that the ratio (0.39) for PARI adopted in this study has been widely
accepted in the previous studies (Smith and Gottlieb; 1974; Hu et al., 2007a). Meanwhile, the values of −2, −1, and −1 W
m$^{-2}$ were used as the lower limits for the PARI, the UVAI, and the UVBI in the "Physically possible limits test", respectively.

### 4.3.2 Extremely rare limits test

A more stringent test is the so-called "Extremely rare limits test", in which a radiation record is further checked to determine
whether it falls outside the interval acquired from historical measurements at the similar time (Ohmura et al., 1998). If a
radiation measurement has successfully passed the "Physical possible limits test" but exceeds the upper/below limit in this
test, the corresponding QC flag of the measurement is set to 4/3, otherwise, the HARDQC procedure goes to the next step.
The lower and upper limits of the "Extremely rare limits test" employed in this study are presented in Table 5.

**Table 5.** Lower and upper limits in the "Extremely rare limits test".

| Lower limit (W m$^{-2}$) | Radiation component | Upper limit (W m$^{-2}$) | Source |
|---|---|---|---|
| −2 | $I_g$ | $I_{sc} \times E_0 \times 1.2 \times \mu^{1.2} + 50$ | Long and Dutton, 2002; |
| −2 | $I_b$ | $I_{sc} \times E_0 \times 0.95 \times \mu^{0.2} + 10$ | CMA, 2007 |
| −2 | $I_d$ | $I_{sc} \times E_0 \times 0.75 \times \mu^{1.2} + 30$ | |
| −2 | $I_r$ | $I_{sc} \times E_0 \times \mu^{1.2} + 50$ | |
| 60 | $I_{dl}$ | 500 | |
| 60 | $I_{ul}$ | 700 | |
| −1 | $I_{par}$ | $0.39 \times I_{sc} \times E_0 \times 1.2 \times \mu^{1.2} + 20$ | This work |
| −1 | $I_{uva}$ | $0.06 \times I_{sc} \times E_0 \times 1.2 \times \mu^{1.2} + 3$ | |
| −1 | $I_{uvb}$ | $0.01 \times I_{sc} \times E_0 \times 1.2 \times \mu^{1.2}$ | |



The limits for the GSWI, the DSWI, the DifSWI, the UpSWI, the DnLWI, and the UpLWI in this work are identical to
those provided by the Long and Dutton (2002) and CMA (2007), while the limits for the PARI, the UVAI, and the UVBI
presented in this study are derived according to the relationships between the solar constants over the spectrum of the PAR,
the UVA, the UVB and the whole solar spectrum.

### 4.3.3 Comparison tests between radiation components

Several investigators (e.g., Ohmura et al., 1998; Michalsky et al., 1999; Augustine et al., 2000; Philipona, 2002; Gupta et al.,
2004; Kratz et al., 2010) pointed out that some stable relationships exist between certain irradiances. For example, the global
solar irradiance directly measured by a pyranometer should be similar to the sum of the simultaneous diffuse sky irradiance
measured by a shaded pyranometer and the vertical part of the direct shortwave irradiance observed by a pyrheliometer. In
practice, the gross differences between these two values often arise from inaccuracies in the solar tracker positioning or in
the shading disk driver. Furthermore, the ultraviolet and PAR irradiance has a spectral correlation to the global solar
irradiance (Hu et al., 2007a, 2007b). On the other hand, the long-wave irradiances are constrained by certain boundaries
defined by functions of air temperature. Consequently, the "Comparison tests between radiation components" takes
advantage of the relationships between the radiation components to examine unreasonable radiation measurements. In this
test, the QC flag is set to 2/1 if the irradiance exceeds the upper/lower limit, otherwise, it is set to 0, which represents the
highest quality of the radiation data. The associated limits of the "Comparison tests between radiation components" in this
study are presented in Table 6.

**Table 6.** Test of comparison between relevant radiation components.

| Radiation component | Relationship | Condition | Source |
|---|---|---|---|
| $I_g; I_b$ | $0.92 \leq I_g/(I_d + I_b) \leq 1.08$<br>$0.85 \leq I_g/(I_d + I_b) \leq 1.15$ | $\theta_z < 75°$ and $I_g > 50\ \mathrm{W\ m^{-2}}$<br>$\theta_z \geq 75°$ and $I_g > 50\ \mathrm{W\ m^{-2}}$ | CMA, 2007 |
| $I_d$ | $(I_d/I_g) < 1.05$<br>$(I_d/I_g) < 1.10$ | $\theta_z < 75°$ and $I_g > 50\ \mathrm{W\ m^{-2}}$<br>$\theta_z \geq 75°$ and $I_g > 50\ \mathrm{W\ m^{-2}}$ | |
| $I_r$ | $I_r < I_g$ | $I_g > 50\ \mathrm{W\ m^{-2}}$ | |
| $I_{dl}$ | $0.4 \times \sigma T_a^4 < I_{dl} < \sigma T_a^4 + 25$ | | |
| $I_{ul}$ | $\sigma(T_a - 15)^4 < I_{ul} < \sigma(T_a + 25)^4$<br>$I_{ul} - 300 < I_{dl} < I_{ul} + 25$ | | |
| $I_{par}$ | $0.3 \leq (I_{par}/I_g) \leq 0.6$ | | Papaioannou et al., 1993<br>Hu et al., 2007a |
| $I_{uva}; I_{uvb}$ | $0.02 \leq (I_{uva} + I_{uvb})/I_g \leq 0.08$ | | Hu et al., 2007b |



# 5 Results

## 5.1 Establishment of the NBSR dataset

### 5.1.1 Process of the NBSR dataset construction

The NBSR dataset is composed of five multi-temporal resolution datasets, i.e., the 1-min (L1B), hourly (L2A), daily (L2B), monthly average hourly (L3A), and monthly average daily (L3B) datasets. Figure 5 shows the flow chart of the generation of the NBSR dataset as follows: At first, the HARDQC approach is applied together with the 1-min raw data of irradiance and meteorological elements to produce the L1B dataset. The L1B dataset contains a header record and six groups of actual data. Secondly, the hourly dataset (L2A) is derived from the L1B dataset by using statistical methods. Note that only the 1-min

radiation data with QC_flag $\leqslant$ 2 rather than the idealistic QC_flag = 0 were selected to calculate the hourly statistics (i.e., hourly mean, minimum, maximum, and standard deviation) , which states a compromise between keeping as much hourly data samples as possible and ensuring that each hourly data has a relatively high quality. Thirdly, the daily dataset (L2B) is derived based on the L2A dataset, in which each daily data is computed in terms of the hourly data when the actual number of hourly data in a day is greater than 90% of its maximum values in a day. At last, the L2A and L2B datasets are taken as

input to yield the monthly average hourly dataset (L3A) and monthly average daily dataset (L3B), respectively.

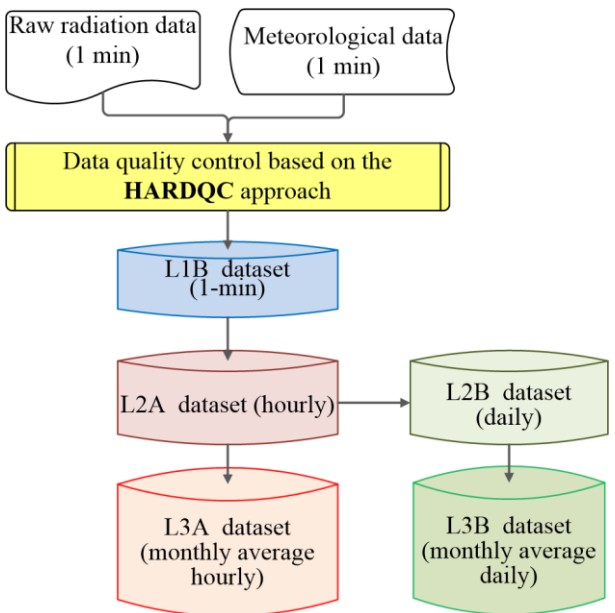

**Figure 5**. Procedure for the establishment of the NBSR datasets.




### 5.1.2 Structure and content of the NBSR dataset

The NBSR dataset is stored as a Hierarchical Data Format (HDF5) file owing to its high data access efficiency and
portability. The HDF5 is a data storage format developed by the National Center for Supercomputing Applications (NCSA,
2019) and is widely applied in astronomy, Earth science, etc. The NBSR dataset contains six groups as follows.

1) *Time*, in which the Year, Month, Day, Hour, and Minute corresponding to the observation data are stored in Beijing
time (BJT=UTC+8), the local standard time (LST), and the true solar time (TST), respectively.

2) *Astronomical variable*, which consists of theoretical calculations of astronomical parameters such as the eccentricity
correction factor of the Earth's orbit, solar zenith angle, solar azimuth angle, etc.

3) *Irradiance variable*, which includes nine fundamental radiation components, i.e. the GSWI, DnSWI, DifSWI,
UpSWI, PAR, UVAI, UVBI, DnLWI, and UpLWI, and the corresponding QC flags. In addition, five additional radiation
components (i.e. the DSWI, PARI, NSRI, NLRI, and the NTRI), which are derived from the fundamental radiation
components, are also included in the *Irradiance variable* group.

4) *Atmospheric variable*, which consists of three variables to depict the sky conditions, i.e., the clear index, the diffuse
coefficient, and the ratio of VUA to UVB.

5) *Surface variable*, in which only one surface variable, i.e., the albedo derived from the UpSWI and the DnSWI, is
available up to now.

6) *Meteorological variable*, which contains simultaneous measurements of six meteorological near surface variables,
i.e., the air pressure, relative humidity, air temperature, visibility, wind speed, and wind direction. These meteorological
variables can provide necessary auxiliary information for analyzing the radiation data.

## 5.2 Assessment of the NBSR dataset

### 5.2.1 Data gaps and data integrity of the NBSR dataset

Figure 6 exhibits the time series of daily irradiance of nine radiation components observed at SDZ during 2013−2022, in
which five apparent data gaps existed. The first data gap occurred in the period from 23 January 2015 to 24 August 2017
(denoted with ① in Fig. 6), in which both the UpSWI and the UpLWI were improperly recorded due to an error setting of
the data logger. This problem was not founded until on 24 August 2017, when the first QC was performed on the raw data of
the NBSR. The second data gap appeared in the period from 6 November to 5 December 2018, in which all instruments were
unloaded and sent to the manufacturer (Jiangsu Radio Scientific Institute Co., Ltd.) to calibrate against the reference
instruments (denoted with ② in Fig. 6). The calibration of instruments essentially improved the reliability of measurements
of the NBSR, but it bought a data gap in data time series in 2018. Due to the failure of the motherboard in the data logger
caused by a lightning stroke on 1 June 2020, the third data gap appeared in the UpSWI and the UpLWI from 1 June to 26
July 2020 (denoted with ③ in Fig. 6). Moreover, the measurements of the DifSWI, the UpSWI, the UpLWI, and the UVBI





were interrupted from 12 May 2021 to 28 June 2021 (denoted with ④ in Fig. 6) because of the malfunction of the
pyranometers (#163082, and #185056), pyrgeometer (#1679), and UV-S-AB-T sensor (#120109). Unfortunately, a light
stroke occurring on 17 July 2021 resulted in a lack of measurements for all radiation components from 17 July to 13
September 2021 (denoted with ⑤ in Fig. 6).

The daily irradiance series reveals that all radiation components have a significant inter-annual variation and periodic
features over the entire observation period (Fig. 6). Compared with the shortwave irradiance, the fluctuation of the long-
wave irradiance (Fig. 6d; 6i) is relatively smooth because the former is prone to be disturbed by the clouds, aerosols, water
vapor, and gases in the atmosphere (Opálková et al., 2019; Wang et al., 2021). It is interesting that the snow cover can be
identified via an abrupt increase in the UpSWI owing to its high albedo that remarkably augment the upwelling shortwave
irradiance (Fig. 6h).

**Figure 6.** Time series of daily GSWI (a), DSWI (b), DifSWI (c), DnLWI (d), PARI (e), UVAI (f), UVBI (g),
UpSWI (h), and UpLWI (i) observed by the NBSR from 1 January 2013 to 31 December 2022.

A data integrity, which is defined as a ratio of the number of actual valid radiation records to the total record number in a year, was introduced to elucidate the completeness of the radiation dataset in this study. Figure 7 indicates that most of the radiation components observed by the NBSR have a data integrity > 90% over the entire period except for eight unusual

cases. For instance, affected by the error setting of the data logger as mentioned above, the UpSWI/UpLWI has three extraordinarily low data integrities occurring in 2015 (0.0%/6.0%), 2016 (0.0%/0.0%), and 2017 (35.9%/36.2%) (denoted with ① − ③ in Fig. 7). Influenced by the instrument calibration performed in 2018 (denoted with ④ in Fig. 7), the data integrities of all radiation components declined to some extent, especially for those of PARI (84.7%) and UVBI (79.7%). In addition, the low data integrities of the PARI (85.2%) and UVBI (88.2%) in 2019 were mainly attributed to the malfunction

of the Li190SB (#46893) and the UV-S-AB-T (#120109) instruments (denoted with ⑤ in Fig. 7). The lightning stroke occurring on 1 June 2020 at SDZ shrank the data integrity of UpSWI/UpLWI to 84.7% and 84.1% in 2020, respectively (denoted with ⑥ in Fig. 7). As a consequence of malfunction of the pyranometers (#163082, and #185056), the pyrgeometer (#1679), and the UV-S-AB-T sensor (#120109) occurring in 2021, the data integrities of DifSWI, UVBI, UpSWI, and UpLWI were 86.3%, 83.3%, 76.2%, and 75.9%, respectively (denoted with ⑦ in Fig. 7). Note that the data integrities of all

radiation components were less than 90% in 2022, primarily affected by the data logger fault rooted in a light stroke occurring on 17 July 2021 at SDZ   (denoted with ⑧ in Fig. 7).

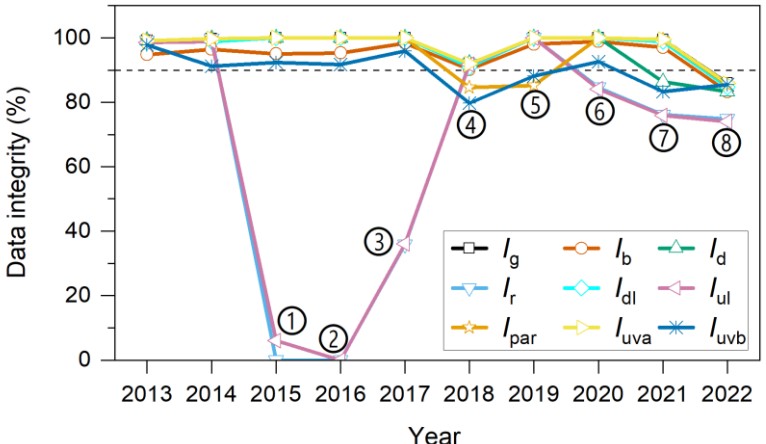

**Figure 7**. Data integrity of nine radiation components at the SDZ during the period 2013–2022.

## 5.2.2 Assessment of the QC of the NBSR dataset

Theoretically, the maximal number of 1-min raw data of the long-wave radiation component is 525 600 in a common year or 527 040 in a leap year, and the number of the shortwave radiation component data is approximately half of that of the former because of the null values recorded for the shortwave radiation during nighttime. However, the number of actual measurements of radiation is less than the theoretical value due to disturbances such as instrument failure, incorrect operation, synoptic events, etc. Figure 8a indicates that the DnLWI was almost continuously observed by the NBSR over the



whole period with a data number varying from 480 148 (in 2018) to 527 035 (in 2016). Nevertheless, the UpLWI suffered serious deficits in 2016, 2015, and 2017, and the corresponding data numbers are 0, 30 866, and 186 643, respectively. The GSWI has been observed nearly undisturbed over the whole period with a data number per year varying from 217 138 (in 2022) to 263 649 (in 2019). Similar to the GSWI, the DifSWI has also been uninterruptedly measured during the entire period with a total number per year from 209 724 (in 2022) to 263 337 (in 2019). Affected by the occasionally inaccurate

solar tracking of the FT-ST22, the actual valid data number of the DSWI ranges from 170 168 (in 2022) to 212 109 (in 2017). In addition, the valid data number of the PARI is from 132 939 (in 2021) to 182 273 (in 2016), and the data number of the UVAI/UVBI ranges from 81 162 (in 2014) to 146 007/144 830 (in 2017). On the other hand, the UpSWI has less data numbers in 2016 (0), 2015 (11 409), and 2017 (79 053) due to the influence of the error setting in the data logger, but more than 185 666 in other years.

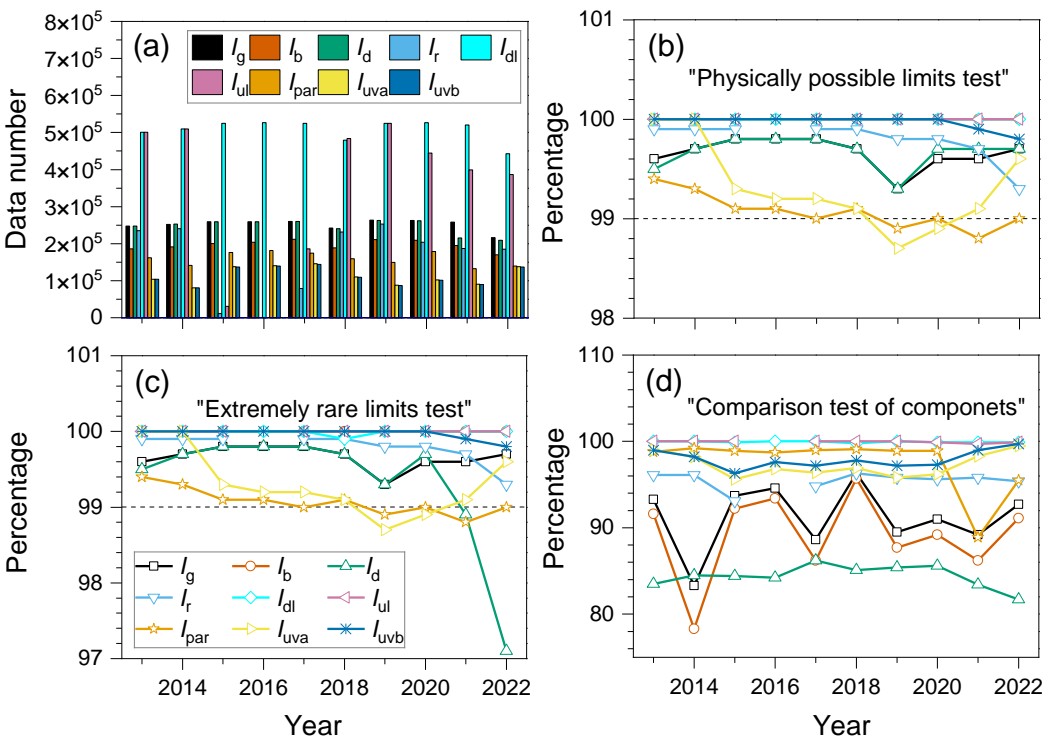


**Figure 8**. Actual data number of the 1-min raw irradiances measured by the NBSR over the period 2013–2022 (a), as well as the percentage that passed the "Physically possible limits test " (b), the "Extremely rare limits test" (c), and the "Comparison test of components" (d).

Figure 8b demonstrates that more than 99.0% of radiation component data have passed the "Physically possible limits

test" in all years except for the UVAI in 2019 (98.7%) and in 2020 (98.9%) as well as the PARI in 2019 (98.9%) and in 2021 (98.8%). By comparing Fig. 8b and Fig. 8c, we found that the 1-min raw data that have passed the "Physically possible limits test" can completely pass through the "Extremely rare limits test" for all radiation components except for the DifSWI in 2021 (98.9%) and in 2022 (97.1%). On the other hand, the "Comparison test of components" can certainly filter out some





radiation data that has passed the previous two tests, i.e., the percentage ranges that passed all three tests of the 1-min raw
data are GSWI (83.3%–96.3%), DSWI (78.3%–95.7%), DifSWI (81.7%–86.2%), UpSWI (93.1%–96.3%), PARI (88.9%–
99.2%), UVAI (95.6%–99.5%), UVBI (96.3%–99.7%), DnLWI (99.8%–100.0%), and UpLWI (99.7%–100.0%),
respectively (Fig. 9d). Note that the percentages of 1-min raw data that passed the "Comparison tests of components"
fluctuated significantly during 2013–2022 for the GSWI and the DSWI, but were relatively stable at a lower level of ~84.4%
for the DifSWI. This phenomenon can be explained by the influence of the imprecise tracking of the sun on the
measurements of the GSWI, the DSWI, and the DifSWI.

## 5.3 Potential application of the NBSR dataset

Though it is not likely to present all possible applications of the high-temporal-resolution NBSR dataset in this study, we try
to give some summaries and prospects on the potential application of the NSRA dataset as follows.

### 5.3.1 Validation of the satellite-derived surface radiation flux and numerical model prediction

High-temporal-resolution (1 min) in situ measurements of radiation components can match the precise timing of a satellite
overpass, and has great advantages in verifying satellite-derived surface radiation flux over different climatic regions
(Ohmura et al., 1998). For instance, Kratz (2010) took advantage of a number of BSRN measurements to verify the Clouds
and the Earth's Radiant Energy System (CERES) Edition 2B products for a wide variety of surface conditions. Yan et al.
(2011) employed the high-temporal-resolution surface radiative flux measurements taken at the SACOL station to evaluate
the performance of the Terra/Aqua Single Scanner Footprint (SSF) product over the Loess Plateau in China. In addition,
Quan et al. (2009b) validated the net surface solar radiation products of CERES/SSF based on the OBSR measurements at
SDZ during January, April, July, and October in 2005. The results show that the SSF products retrieved from the Li model
(Li et al., 1993) and the Masuda model (Masuda et al., 1995) overestimate the net surface flux by 62.2 W m$^{-2}$ and 50.8 W
m$^{-2}$ under clear sky conditions, respectively. As one of the crucial observation of the SDZ regional GAW station, the NBSR
dataset is expected to play a more vital role in the interpretation of the interaction between radiation and atmospheric
components, validation of satellite-derived surface radiation flux (e.g., the CERES flux products, the Fengyun-4
Geostationary satellite-based solar energy products, etc.) as well as in the satellite-based solar energy nowcasting (Huang et
al., 2022).

Owing to its accurate measurement of the radiation fluxes, the high-temporal-resolution radiation measurement from
radiation observation networks like the BSRN has been widely used to assess and improve the performance of general
circulation models (GCMs) and radiation transfer models (e.g., Ohmura et al., 1998; Wild et al., 2006, 2019). For example,
the BSRN data are useful for the interpretation of the underestimation of atmospheric long-wave emission as well as the
clarification of the debate regarding the missing shortwave absorption in the atmosphere in the GCMs (Wild et al., 1995,
1996). In addition, Liu et al. (2008) evaluated the clear-sky net surface solar radiation calculated by the Moderate Resolution




Transmittance Model (Berk et al., 1999) based on a 1-min dataset of DnSWI and UpSWI observed by the OBSR at SDZ in
2005. As a successor and improved version of the OBSR, the NBSR is expected to play a significant role in investigation of
radiation codes, climate models, and numerical weather prediction models.

### 5.3.2 Investigation of diurnal cycles of the radiation components

The high-temporal-resolution radiation measurements are essential to accurately trace the diurnal cycle, which is important
in GCM simulations (Dutton, 1990). Figure 9 shows the diurnal variations of the nine radiation components measured at
SDZ. The annual hourly mean value and the associated standard deviation for each radiation component were calculated in
terms of the hourly data at the same hour measured in 2013. The reason why to choose the L2A dataset in 2013 as a data
source for calculating the annual hourly mean value and standard deviation attributes to its highest data integrity and quality
in this year. The GSWI, DSWI, DifSWI, and the UpSWI display the familiar bell shape, rising to a peak during midday (~
1300 BJT) and possess a zero value during the night (Fig. 9a). The diurnal course of the GSWI, DSWI, and DifSWI is
caused by the diurnal cycle of the solar zenith angle, but also varies with atmospheric composition and cloudiness. The
UpSWI cycle is dominated by the GSWI cycle but is also influenced by the surface albedo, which varies with vegetation and
growing season, surface modification, moisture, snow cover, and incident angle of the sun (Dutton, 1990). Furthermore, the
standard deviation of the hourly irradiance is larger at noon while smaller at sunrise and sunset, which may be impacted by
its absolute values as well as the diurnal variation of the water vapour in the atmosphere, i.e., the moisture in the atmosphere
is generally lower in early morning or dusk but reaches its maximum at noon (Hu et al., 2007a). The diurnal cycle of the
DifSWI, which is useful for investigating atmospheric physics and for evaluating the implications for biological systems, is
affected by the variation of the GSWI as well as the atmospheric conditions such as the aerosol loading (Che et al., 2005).

As a part of the solar radiation in the visible spectrum (400−700 nm), the diurnal variation of the PARI is closer to that
of the GSWI in spite of the ratio of the PARI to GSWI would be modulated by the water vapour in the atmosphere (Hu et al.,
2007a). The hourly PARI at SDZ reaches its maximum around noon with a value of 220.3 W m$^{-2}$ and has a minimum of zero
during night time (Fig. 9b). Ultraviolet radiation is defined as a portion of solar irradiance within the spectrum between 100
and 400 nm. The UVC (100–280 nm) is absorbed by the atmospheric ozone, whereas most radiation in the UVA (315–400
nm) and about 10% of the UVB rays (280–315 nm) reach the Earth's surface (Hu et al., 2007b). Therefore, the ultraviolet
radiation reaching the Earth's surface is largely composed of the UVA with a small UVB component. The value of UVBI at
the Earth's surface is generally small due to the attenuation of the strong stratospheric ozone absorption, Rayleigh scattering,
etc. However, the UVBI also exhibits the analogous diurnal cycle as the one of the UVAI at SDZ, i.e., the highest values of
the UVAI (31.9 W m$^{-2}$) and the UVBI (0.5 W m$^{-2}$) occur when the sun elevation reaches its maximum (~ 1300 BJT), and
the lowest values appear during the night. Note that the ratio of the UVAI to UVBI at SDZ varies from 79.2 at noon to 95.5
at sunrise and sunset with a daily mean value of 84.5 (Fig. 9c).

Figure 9d indicates that the diurnal variation of the hourly mean DnLWI is relatively stable with an average value of
300.0 ±7.1 W m$^{-2}$ over a whole day. Whereas, the standard deviation of the DnLWI is relatively large (~73.3 W m$^{-2}$), which

is affected by the vertically integrated atmospheric temperature, water vapour, ozone, carbon dioxide, and other trace constituents (Dutton, 1990). Moreover, the phase of the diurnal cycle of the UpLWI is postponed an hour compared to the

one of the downwelling shortwave irradiance, e.g., the UpLWI reaches its peak at 1400 BJT rather than at 1300 BJT, which is predominately caused by the large thermal inertia of the land surface (Fig. 9d). On the other hand, the hourly mean UpLWI exhibits a remarkable diurnal cycle with an average value of 365.8 $\pm$ 35.2 W m$^{-2}$, and the standard deviation of the hourly UpLWI ranging from 58.4 to 82.7 W m$^{-2}$ (Fig. 9d).

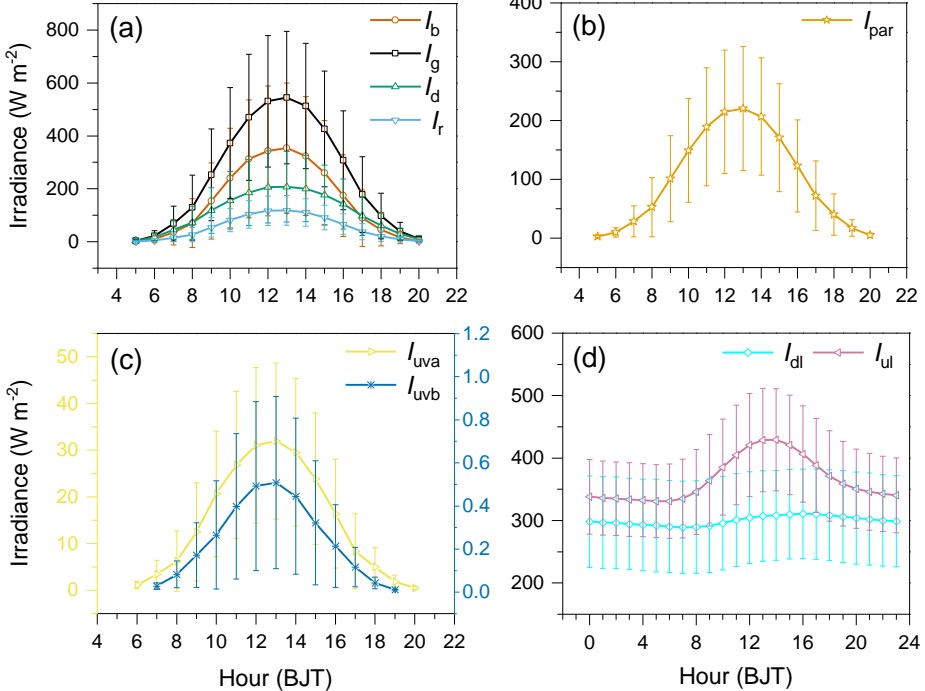

**Figure 9.** Annual averaged diurnal variation in the GSWI, DSWI, DifSWI, and UpSWI (a), the PARI (b),
the UVAI and UVBI (c), and the DnLWI and UpLWI (d) at SDZ. The vertical lines denote the standard
deviations calculated from the hourly irradiances in 2013.

### 5.3.3 Acting as a reference to correct other analogous instruments

Using the parallel sunshine duration (SD) observations from the Jordan sunshine recorder and the DFC2 photoelectric

sunshine meter as well as those derived from the measurements of the CHP1 pyrheliometer and CMP11 pyranometer of the NBSR, we have elucidated the mechanism that produced inconsistency among these instruments and proposed a simple linear regression function to improve the consistency of the long-term SD series observed at SDZ (Zhou et al., 2021). Hence, the highly accurate radiometric instruments of the NBSR can act as a reference to improve the consistency of long-term SD records measured by different SD recorders, which may be beneficial for climate studies as well as for the promotion of



tourist destinations since the SD takes into account the psychological effects of strong solar light on human well-being (WMO, 2006).

### 5.3.4 Establishing parameterization models for retrieving surface radiation components

As pointed out by Ohmura et al. (1998), the measurements at the BSRN stations have provided detailed information of radiative effects on the climate system and contributed to a better parameterization of radiation in climate models. In our
recent research, a long-term (2011−2022) hourly dataset of DnLWI and associated meteorological elements obtained from seven CBSRN stations including the SDZ were used to recalculate the coefficients of the Brunt model (Brunt, 1932) and the Weng model (Weng et al., 1993) as well as to develop a new parameterization model to estimate the downwelling long-wave irradiance at the surface (Yang et al., 2023).

### 5.3.5 Assistance to evaluate the effects of aerosol loading on SD

The clear-sky hourly clearness index ($K_t$), which is defined as the ratio of the measured hourly global shortwave irradiance to the corresponding extra-terrestrial irradiance at the top of the atmosphere under cloud-free conditions, can be taken as an indicator to represent the atmospheric aerosol loading (Iqbal, 1983; Che et al., 2007). In our previous study, we found that the start time of SD at SDZ is 0800, 0600, and 0500 BJT under high ($0.70 \leqslant K_t < 1.0$), middle ($0.55 \leqslant K_t < 0.7$), and low ($K_t < 0.55$) levels of clearness index, which were calculated in terms of the GSWI observed by the NBSR, respectively. The end
time of hourly SD under the middle/high clearness index is one/two hours earlier than that at low clearness index, which may be partly influenced by the blocking effect of hills located to the west of the SDZ (Quan et al., 2023). The value of hourly SD in the second period (1000−1600 BJT) under high aerosol loading (or low level of clearness index) is ~ 0.6 h h$^{-1}$, i.e., approximately 60% and 56% of that under middle and low aerosol loadings, respectively (Quan et al., 2023).

## 6 Summary and suggestions

In this study, the NBSR at the SDZ was introduced in detail for the first time. As an unique atmospheric background station in North China, SDZ has a good spatial representativeness of these area. Therefore, the measurements at the SDZ including the high-temporal-resolution radiation data are expected to contribute to great achievements in a lot of scientific fields in the future. The objective of this study includes the introduction of the site and instruments, the development of the HARDQC algorithm, the establishment and assessment of the NBSR dataset, as well as some demonstrations on its potential
applications. The major conclusions of this study are as follows.

The HARDQC algorithm presented in this study proved to be effective in quality controlling the 1-min raw data of the nine radiation components observed by the NBSR. More than 99.0% of 1-min raw data of the nine radiation components except for the UVAI (98.7%) and the PARI (98.9%) have passed the "Physically possible limits" test over the period 2013–



2022. Apart from the DnLWI (97.1%), the percentages of other radiation components that have passed the "Extremely rare limits" test are greater than 98.6%. Nevertheless, the percentages that have passed the "Comparison tests between radiation components" over the period 2013–2022 range from 83.3% to 96.3% for the GSWI, for the DSWI (78.3% – 95.7%), the DifSWI (81.7% – 86.2%), the UpSWI (93.1% – 96.3%), the PARI (88.9% – 99.2%), the UVAI (95.6% – 99.5%), the UVBI (96.3% – 99.7%), the DnLWI (> 99.8%), and the UpLWI (> 99.7%), respectively.

Adverse synoptic conditions (e.g., lightning stroke), improper operation, and instrument removal for calibration are three primary reasons leading to the data deficiency of the NBSR. Therefore, it is highly recommended that a backup radiation observing system should be simultaneously operated with the NBSR in order to solve the issues of unpredictable instrument failures and missing data.

All instruments in the NBSR except for the UV-S-AB-T and Li190SB are considerably stable with a relative deviation less than 1% over the whole observation period according to the results of instrument calibration. Thus, it is recommended that the UV-S-AB-T and the Li190SB should be calibrated regularly (e.g. once per year) in future operations. The IR02 pyrgeometer is found to output abnormal long-wave irradiance in the case of extreme cold and dry conditions due to its large temperature dependency. We, thus, suggest that the IR02 pyrgeometer should be replaced by a more precise instrument (e.g., the CGR4) in the future.

The representativeness of the UpSWI and the UpLWI is insufficient because the face-down CMP11 and IR02 are mounted on the top of the 1.5 m high steel bracket on the MOF rather than installed on a 10 m tall tower. By means of the high tower, the face-down radiometric instruments can view a wider underlying surface, which could be very helpful to improve its representativeness of the actual underlying surface.

Despite the existence of a few imperfections in the NBSR dataset, the long-term (more than ten years) quality-assured dataset of nine radiation components can provide surface radiation budget data for many applications such as the validation of satellite products and numerical models, investigations on the interaction between radiation fluxes and atmospheric composition, and the detection of the long-term changes in the radiative fluxes, etc. Moreover, it is very useful for interpreting the physical and chemical processes along with other atmospheric component observations at the SDZ station.

In the near future, the dataset of the OBSR is also planned to be established to prolong the NBSR dataset back in time up to more than eighteen years. Both the OBSR dataset and the NBSR dataset are designed to be permanently archived and freely accessible, which will contribute to radiation-related scientific studies in the future.

**Data availability.** The daily dataset of the NBSR (2013−2022) presented in this study is available online at https://issues.pangaea.de/browse/PDI-35544. The ".csv" file can be opened in any text editor or spreadsheet program (e.g., Excel). In addition, the hourly radiation dataset for 2016 can be downloaded from https://www.doi.org/10.12072/SDZRAW.001.2017.db (Quan and Wang, 2017), and the hourly radiation dataset for 2017 is available from https://www.doi.org/10.12072/SDZRAW.003.2018.db (Quan and Wang, 2018).



**Author contributions.** WQ contributed the ideas, organized the research, edited the manuscript, and provided the funding

acquisitions; ZW, YL, and XY performed the data processing such as the QC procedure on the radiation components; LQ

and XZ contributed to an analysis of the data and calibration of the instruments; JJ and ZM provided the instrument

maintenance and data collection; MW contributed to improve the ideas, to check and rectify the manuscript.

**Competing interests.** The contact author has declared that neither him nor his co-authors have any conflict of interest.

**Acknowledgements.** This study is funded by the China Scholarship Council (No. 202205330024), National Key Research

and Development Program of China (Grant No. 2017YFB0504002), National Natural Science Foundation of China (Grant

No.42275199), National Science and Technology Infrastructure Platform Project (2017), and the Special Fund for Basic

Scientific Research of Institute of Urban Meteorology (Grant No. IUMKY201735). We would like to thank Mr. Yao Wang

of the Beijing Meteorological Observatory for providing the drone photo of Shangdianzi station and Prof. Bo Hu of the

Institute of Atmospheric Physics, Chinese Academy of Science for his cherished suggestions on how to use the PAR data.

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
