# Peer review of "A quality-assured dataset of nine radiation components observed at the Shangdianzi regional GAW station in China (2013–2022)"

_Earth System Science Data, 2023_

## Author Comment (AC1)

**Response to Reviewers**

**A quality-assured dataset of nine radiation components observed at the Shangdianzi regional GAW station in China (2013−2022)**

Weijun Quan, Zhenfa Wang, Lin Qiao, Xiangdong Zheng, Junli Jin,
Yinruo Li, Xiaomei Yin, Zhiqiang Ma, and Martin Wild

22 December 2023

Dear editor and reviewers,

We would like to thank the editor for handling our manuscript and the reviewers for their careful evaluation of our work and the valuable comments, suggestions, and questions. We believe that the manuscript will considerably benefit from the reviewers' feedback. Our detailed responses to all the comments are addressed as follows.

In order to clearly address each of the comments, we have copied comments in blue font and have addressed them one by one in black font. In addition, we use black italic font to quote text from the revised manuscript.

Yours sincerely,

Weijun Quan, Martin Wild, and co-authors

**Response to Reviewer 1 Comments**

**General comments:**

This study established and evaluated a quality-assured and relatively long-term dataset (2013-2022) comprising nine radiation components observed at the Shangdianzi regional Global Atmosphere Watch (GAW) station. This dataset serves as a crucial foundation for investigating solar radiation variations, validating satellite products, and model simulations, among other applications. The manuscript detailly outlines the quality control procedures, assesses data integrity, and provides explanations for data gaps. Additionally, a concise analysis of diurnal variations is included, demonstrating clear logic and well-organized content. While this study is of utmost significance, a few minor concerns should be addressed, as detailed below:

Thank you for this comment. We are grateful for reviewer's constructive comments.

**Specific comments:**

1. Although the quality control procedures have encompassed essential tests, such as the physically possible limits test, extremely rare limits test, and comparison test between

radiation components, it is worth considering additional challenges encountered when measuring solar radiation, such as addressing zero offsets in pyranometers and conducting tracker-off tests. Providing insights into how these issues were addressed would enhance the paper's completeness.

Thank you for this valuable comment. It is necessary to address the additional information in the manuscript on how to solve the challenges such as the zero offset in pyranometers and solar-tracking precision. We have consulted the manufacture of the NBSR and have been informed that the zero offset in pyranometers is generally alleviated through adding a ventilation system for the pyranometers during observation. Furthermore, a built-in program in the data collector is applied to automatically correct the zero offset in pyranometer and the solar tracing position.

To make this clear, we have modified the sentence in section 4.3 (L314 – L316) as: "Though the NBSR is well maintained and all instruments are regularly calibrated *as well as the zero offsets in pyranometers and solar tracking precision are automatically amended via a built-in program in the data collector*, some irrational records still exist due to the influence of adverse weather, operational mistakes, power failure, data transmission interrupt, etc."

2. Regarding the calculation of monthly average solar radiation from daily values, two specific aspects require clarification: (1) handling data gaps in the daily time series and (2) establishing a threshold for the ratio of daily observations to account for days in a month. Elaborating on these aspects, including strategies for addressing data gaps and specifying the threshold criteria, would improve transparency.

Thank you for your reminder.

We have modified the sentence in section 5.1.1 (L394 – L395) as: "At last, the L2A and L2B datasets are taken as input to yield the monthly average hourly dataset (L3A) and monthly average daily dataset (L3B) *when the number of valid L2A and L2B files in a month is greater than 25 (Wang et al., 2007)*, respectively." In addition, a reference (Wang et al., 2007) is also added to the reference list.

3. The omission of annual average solar radiation and an in-depth analysis of changes from 2013 to 2022 is a notable gap. To enhance the study, it is recommended to extend the analysis by incorporating interannual variation and trend analysis. This addition would provide a more comprehensive understanding of the dataset's temporal dynamics.

Thank you for your valuable recommendation.

We have expanded the temporal variation of the radiation components (Section 5.3.2) and supplied a figure (Fig. 10) and a paragraph to elaborate the interannual variation and trend analysis of solar radiation as following:

*"Figure 10 shows the inter-annual variations and associated linear trends for three shortwave radiation components (i.e., the GSWI, DSWI, and DifSWI) over the SDZ in the period 2013–2021. As a prerequisite in constructing a time series of annual irradiance, a few missing monthly irradiances, e.g., the monthly GSWI, DSWI, and DifSWI in November 2018 as well as the DifSWI in May and June 2021, were replaced by the averages of the corresponding months over all other years during the 2013–2021. Note that the annual radiation data of 2022 were not selected because too many missing observations (more than three months) appeared in this year, which leads the annual radiation data of the year 2022 to be highly suspicious. The annual day-time GSWI (with a linear trend of 21.2 W m$^{-2}$/10a) and DSWI (with a linear trend of 23.9 W m$^{-2}$/10a) increase over the period 2013–2021, while the annual day-time DifSWI decreases with a linear trend of –4.9 W m$^{-2}$/10a. A previous study indicated that both the total cloud cover (with a linear trends of –0.8 tenth decade$^{-1}$) and the low cloud cover (with a linear trends of –1.2 tenth decade$^{-1}$) over the SDZ suffered remarkable decreases in the 2010s (Quan et al., 2023). In general, a decrease in cloud cover will increase the total and direct solar radiation reaching the ground as well as alter the ratio between the DSWI and DifSWI (Wild et al., 2019). Thereby, it is reasonable to believe that increasing trends of shortwave radiation over the SDZ are mostly attributed to the decline of the cloud cover. On the other hand, the linear trends of the GSWI, DSWI, and DifSWI are also influenced by the variation of aerosol loading over the SDZ. For instance, the surface particulate matter with an aerodynamic diameter of < 2.5 μm (PM$_{2.5}$) measured at the SDZ reached its maximum value in 2013 and declined subsequently after 2014, which is attributed to the severe haze episodes occurred over the BTH in 2013 and the implementation of the air pollution action plan at the end of 2013 (Fu et al., 2020). Moreover, the relationship between the DSWI and DifSWI can be modulated by the particulates in the atmosphere due to their scattering effects on solar radiation. It is worth noting that the dramatic declines in the GSWI and the DSWI as well as the incline in the DifSWI in 2021 were essentially caused by an increase of precipitation clouds in 2021, in which the historic maximum precipitation amount (1047.3 mm) and 3rd highest number of rainy days (108 days) over the past six decades occurred at the SDZ (Quan et al., 2023)."*

[Figure]

*Figure 10.* *Inter-annual variations of the annual mean day-time GSWI, DSWI, and DifSWI and associated linear trends for GSWI (denoted with a thick black dashed line), DSWI (denoted with a thick red line), and DifSWI (denoted with a thick green dashed line) over SDZ area during the period 2013–2021, respectively.*

---

## Author Comment (AC2)

**Response to Reviewers**

**A quality-assured dataset of nine radiation components observed at the Shangdianzi regional GAW station in China (2013−2022)**

Weijun Quan, Zhenfa Wang, Lin Qiao, Xiangdong Zheng, Junli Jin,
Yinruo Li, Xiaomei Yin, Zhiqiang Ma, and Martin Wild

22 December 2023

Dear editor and reviewers,

We would like to thank the editor for handling our manuscript and the reviewers for their careful evaluation of our work and the valuable comments, suggestions, and questions. We believe that the manuscript will considerably benefit from the reviewers' feedback. Our detailed responses to all the comments are addressed as follows.

In order to clearly address each of the comments, we have copied comments in blue font and have addressed them one by one in black font. In addition, we use black italic font to quote text from the revised manuscript.

Yours sincerely,

Weijun Quan, Martin Wild, and co-authors

**Response to Reviewer 2 Comments**

**General comments:**

This manuscript reported 10 years of measurement of nine radiation components at the Shangdianzi station in China. The important details regarding measurement and quality control are well explained, making this dataset of great value to the community. The manuscript is well written and potential application of the dataset is discussed. I recommend publication of this work and I only have minor comments.

Thank you for this comment. We are grateful for the reviewer's constructive comments.

**Specific comments:**

1. Section 3.2. The instruments for several radiation components have changed over the years, such as for Inb, Id. Were any cross-validation or instrument performance comparisons done for these instruments? Instruments with the same model and manufacturer can sometimes behave differently, so it is important to make the comparison.

Thank you for this valuable comment.

It is really important to compare the instruments used in our work against the reference ones to improve the consistency of radiation measurements over a long period. To this end, we sent these instruments to the manufacturer (Jiangsu Radio Scientific Institute Co., Ltd.) to compare against the reference instruments (e.g. the CM21 pyranometer, the CHP1 pyrheliometer, the CGR4 pyrgeometer, the UVS-AB-T radiometer, and the Li-200190SB sensor). These reference instruments had been compared against the national radiometric standards of China (e.g. the CM22 pyranometer, the H-F absolute cavity radiometer, the CG4 pyrgeometer, etc.), which were transferred from the World Radiation Center in Davos, Switzerland (e.g., Quan et al., 2010; Yang et al., 2015; PMOD/WRC, 2022; Yang et al., 2023).

2. Section 3.2.2. What is the frequency of instrument maintenance and calibration? I only see each instrument being calibrated once at the manufacturer and then they are used for several years. Is this frequency sufficient for high-quality measurement?

Thank you for this key question.

- During the period of 2013–2019, two campaigns for instrument calibration had been performed. One was carried out by the manufacture before all these instruments were installed to start observing (approximately Jun 2012). The other was performed in November 2018, in which all instruments used at SDZ were uninstalled and sent to the manufacture to compare with the reference instruments.
- In 2020, we purchased a set of instruments as the proxy for the original operation instruments. Thereby, since 2020, all instruments have been calibrating once a year.
- Fortunately, we found that most of the instruments used in this study, which are manufactured by the Kipp & zonen, are very reliable, i.e., the changes of the instrument's sensitivity are very small even after several year operation (See Table 3). Whereas, it is a pity to loss about one month radiation measurement in November 2018 because we have to send them to calibrate but we cannot provide the proxy instruments.